# Ischaemic Stroke of the “Hand-Knob” Area Due to Paradoxical Cerebral Air Embolism after Central Venous Catheterization—A Doubly Rare Occurrence: A Case Report and an Overview of Pathophysiology, Diagnosis, and Treatment

**DOI:** 10.3390/brainsci12060772

**Published:** 2022-06-13

**Authors:** Paola Nicolini, Andrea Arighi, Elisa Gherbesi, Francesco Maria Lo Russo, Clara Mandelli, Giuseppina Schinco, Stefano Carugo, Tiziano Lucchi

**Affiliations:** 1Geriatric Unit, Fondazione IRCCS Ca’ Granda Ospedale Maggiore Policlinico, 20122 Milan, Italy; clara.mandelli@policlinico.mi.it (C.M.); giuseppina.schinco@policlinico.mi.it (G.S.); tiziano.lucchi@policlinico.mi.it (T.L.); 2Neurodegenerative Disease Unit, Fondazione IRCCS Ca’ Granda Ospedale Maggiore Policlinico, 20122 Milan, Italy; andrea.arighi@policlinico.mi.it; 3Cardiovascular Disease Unit, Fondazione IRCCS Ca’ Granda Ospedale Maggiore Policlinico, 20122 Milan, Italy; elisa.gherbesi@policlinico.mi.it (E.G.); stefano.carugo@unimi.it (S.C.); 4Department of Neuroradiology, Fondazione IRCCS Ca’ Granda Ospedale Maggiore Policlinico, 20122 Milan, Italy; francesco.lorusso@policlinico.mi.it; 5Department of Clinical Sciences and Community Health, University of Milan, 20122 Milan, Italy

**Keywords:** central venous catheterization, cerebral air embolism, hand-knob, iatrogenic stroke, ischaemic stroke, paradoxical embolization, patent foramen ovale

## Abstract

Central venous catheters (CVCs) are increasingly used across specialties for invasive haemodynamic monitoring and for the delivery of fluids, medications, and nutritional support. Cerebral air embolism (CAE) is a rare but potentially fatal complication associated with the insertion, maintenance, and removal of CVCs. It can occur through different mechanisms, including the direct retrograde ascension of air into the cerebral veins and paradoxical embolism due to a right-to-left intracardiac or intrapulmonary shunt. The “hand-knob” area is the cortical region within the primary motor cortex that contains the representation of the hand. It is located in the superior precentral gyrus and is the site of less than 1% of all ischaemic strokes. We report here the case of a patient who experienced an ischaemic stroke of the right “hand-knob” area, due to paradoxical CAE through a previously undiagnosed patent foramen ovale (PFO), after the insertion of a catheter in the right internal jugular vein. We also provide an overview of the pathophysiology, diagnosis, and treatment of CAE. Suspecting CAE in the case of an acute neurological event occurring in close temporal relationship with central venous catheterization is paramount to allow the early recognition and treatment of this uncommon form of iatrogenic stroke.

## 1. Introduction

Central venous catheters (CVCs) are increasingly used across specialties for invasive haemodynamic monitoring and for the delivery of fluids, medications, and nutritional support [1]. The number of CVCs inserted annually has been estimated to be over 5 million in the US [2], over 600,000 in Italy [3], and over 120,000 in the UK’s critical-care environment [4]. The total burden of CVC use has been found to be higher in general medical wards than in intensive-care units [5], raising concerns about a lower awareness of CVC-related complications in less-specialized settings [6].

Air embolism is a rare but potentially fatal complication of the insertion, maintenance, and removal of CVCs, with a prevalence of around 0.1% and a mortality rate of about 20% [1]. Cerebral air embolism (CAE) is more uncommon than pulmonary air embolism and can occur through different mechanisms [1,7]. Venous CAE is due to the retrograde ascension of air into the jugular veins when the patient is in a sitting position [1,7]. Arterial CAE tends to be more likely in the supine position and is caused by the migration of air from the venous to the arterial side of the circulation via a right-to-left shunt [1,7]. Such paradoxical embolism can be associated with intracardiac shunts (i.e., patent foramen ovale (PFO) and atrial/ventricular septal defects) or intrapulmonary shunts (i.e., pulmonary arteriovenous malformations and inducible intrapulmonary arteriovenous anastomoses) [1,7]. The transpulmonary passage of air can take place even in the absence of anatomic shunts when the filtering capacity of the pulmonary capillaries is overwhelmed by large air volumes or anaesthetic agents [1,7].

The most frequent cause of paradoxical CAE is PFO [1,7,8]. PFO is a congenital heart defect which has an approximately 25% prevalence in the general population and allows transient right-to-left interatrial flow through a one-way flap mechanism when the pressure in the right atrium exceeds that in the left atrium [9]. Since right atrial pressure (RAP) is usually lower than left atrial pressure, a PFO is functionally closed most of the time [9]. However, it can open when there is a reversal of the normal left-to-right pressure gradient. This happens under physiological circumstances in the early ventricular systole and after the release of the Valsalva manoeuvre, as well as in pathological conditions such as pulmonary hypertension [9].

The clinical manifestations of CAE are heterogeneous both in nature and severity, and consist of focal and non-focal neurological signs and symptoms [7,10,11,12,13]. The former comprise motor deficits (ranging from monoparesis to tetraplegia, with hemiparesis being the most common) along with hemianopsia, aphasia, and dysarthria [10,13]. The latter include loss or alteration of consciousness and seizures [7,10,11,12,13]. The outcome is often dismal because, besides being potentially fatal, CAE is associated with neurological sequelae in about half of the survivors [1,7,10].

The “hand-knob” area is the cortical region within the primary motor cortex that contains the representation of the hand [14]. It is located in the superior part of the precentral gyrus and is a knob-like structure which protrudes into the central sulcus [14]. It has an inverted omega (90%) or horizontal epsilon (10%) shape on axial neuroimaging and it looks like a posteriorly-directed hook on sagittal neuroimaging [14]. Stroke of the “hand-knob” area represents less than 1% of all ischaemic strokes [15,16,17]. It manifests as an isolated hand paresis and can be easily misdiagnosed for a lesion of the peripheral nervous system; thus, it has also been termed “pseudoperipheral palsy” [15,17]. It has been suggested to have a mainly embolic origin, from cardiac or large artery disease, and it has been shown to have a benign clinical course with no or mild residual deficits [16,17,18,19,20].

We report here the case of a patient who experienced an ischaemic stroke of the right “hand-knob” area, due to paradoxical CAE through a previously undiagnosed PFO, after the insertion of a catheter in the right internal jugular vein. We also provide an overview of the pathophysiology, diagnosis, and treatment of CAE. To the best of our knowledge, there are no previous reports in the literature describing an ischaemic stroke of the “hand-knob” area secondary to paradoxical air embolism from a CVC.

## 2. Case Presentation

An 83 year-old man was admitted to our geriatric ward for elevated liver enzymes. He was a non-smoker and non-diabetic with a past, but not current, history of hypertension and dyslipidaemia. One year earlier, he had been hospitalized for non-resolving pneumonia treated with multiple courses of antibiotics and systemic high-dose steroids. He had been concurrently diagnosed with epiglottic paralysis which had required the placement of a percutaneous endoscopic gastrostomy tube. His medications included low-dose prednisone, esomeprazole, triazolam, and vitamin B/D supplements. His other blood tests were normal, except for mild hyponatriaemia due to excessive water intake for which he was placed on water restriction.

Screening for hepatotropic viruses was negative. Abdominal ultrasound and computed tomography (CT) were unremarkable. He was scheduled for a transcutaneous liver biopsy and, the day before the procedure, he lost his peripheral venous line. Because of poor peripheral venous access, a triple lumen catheter was inserted in the right internal jugular vein using the Seldinger technique under ultrasound guidance, with the patient in the Trendelenburg position.

Within a few minutes from the end of the manoeuvre, he became briefly unconscious and, on regaining consciousness, complained of left upper-limb weakness. His neurological examination was normal and his vital signs were unchanged from previous assessments (blood pressure of 110/60 mmHg, heart rate of 50 bpm, oxygen saturation of 97% on room air). He was therefore maintained under observation. One hour later, he reported sudden-onset weakness of the left hand with wrist drop. Neurological examination showed an isolated flaccid paresis of the left hand. Although he complained of hand numbness, there was no objective evidence of sensory deficits. He was administered 100% oxygen and was referred for an urgent non-contrast brain CT scan. The latter was performed 45 min after the onset of hand paresis, and it excluded haemorrhage, early ischaemic changes, and other possible causes of stroke (Figure 1a). The patient was started on antiplatelet treatment (aspirin 100 mg/daily) and the planned liver biopsy was suspended. Non-contrast brain magnetic resonance imaging (MRI) was obtained the next day, about 22 h after the onset of hand paresis. It showed a recent ischaemic lesion of the right superior precentral gyrus (Figure 1b–d). Both CT and MRI displayed diffuse chronic cerebrovascular disease (Figure 1e–h).

A complete stroke work-up was subsequently performed. Carotid ultrasound revealed only small (<30%) fibrocalcific plaques of the internal and external carotid arteries. On a seven-day continuous electrocardiographic monitoring, there was evidence of sinus bradycardia (mean heart rate of 50 bpm) with no pathological pauses or significant tachyarrhytmias. On transthoracic echocardiography, the ventricular septum was mildly hypertrophic, the left atrium was moderately enlarged, there was mild-to-moderate tricuspid regurgitation with an increased systolic pulmonary arterial pressure (45 mmHg), but no visible intracardiac shunt. Based on the close temporal relationship between the development of neurological dysfunction and central venous catheterization, there was a strong suspicion of paradoxical embolism, and more specific investigations were ordered. Transcranial Doppler ultrasound with agitated saline was negative at rest, but, during a Valsalva manoeuvre, it documented a “shower” appearance of microembolic signals over the middle cerebral artery, indicating a high-grade right-to-left shunt (Figure 2). Transthoracic ecocardiography with agitated saline achieved complete opacification of the right heart at rest; during a Valsalva manoeuvre, it visualized more than 20 bubbles in the left heart chambers within three cardiac cycles from complete opacification of the right atrium (Figure 3, Appendix A), establishing the diagnosis of a high-grade right-to-left intracardiac shunt. The intermittent nature of the right-to-left shunt was consistent with a PFO.

After a joint neurology and cardiology consultation, the patient was continued on low-dose aspirin because, in subjects over 60 years of age, transcatheter PFO closure is not currently recommended [21]. The patient’s left-hand paresis gradually improved and, six days after the onset of stroke, he had undergone full neurological recovery. His liver biopsy was rescheduled during the hospital stay and was performed by a transjugular approach, since it was not possible to discontinue antiplatelet treatment. There were no procedural complications. A histological diagnosis of autoimmune hepatitis was made and his steroid dose was increased. One month after discharge, his liver enzymes had almost normalized and his left-hand motor function had remained persistently normal.

## 3. Discussion

The occurrence of paradoxical CAE generally requires three conditions: venous air embolism, an anatomic defect that allows communication between the right and left heart chambers, and a right-to-left pressure gradient.

The precise cause of venous air embolism in our case remains unidentified. Air embolism has been reported in the literature even when catheterization appears to be conducted according to best-practice guidelines [8], and a recent review has found specific reference to procedure-related errors only in a minority of cases of iatrogenic CAE [13]. It may be speculated that our patient inadvertently took a deep breath during jugular vein cannulation and that he was somewhat dehydrated. Both inspiration and hypovolaemia reduce central venous pressure and facilitate the entry of air into the venous system [1,7]. The iatrogenic character of air embolism emphasizes the importance of prevention by meticulous compliance with cautionary measures like optimizing volume status, Trendelenburg positioning, needle hub/catheter occlusion and appropriate breathing instructions [7,8,10].

As to the reason for the right-to-left shunting, on standard echocardiography our patient had mild pulmonary hypertension (<50 mmHg), possibly as a consequence of his past pulmonary disease, but no intracardiac shunt. Moreover, transcranial Doppler ultrasound and transthoracic echocardiography with agitated saline were negative at rest and became positive only during a Valsalva manoeuvre. These findings suggest that greater RAP was likely needed for shunting through the PFO. It is well-recognized that, in the context of air embolism, RAP can be temporarily elevated by a transitory “air-lock” phenomenon within the right ventricular outflow tract or by simultaneous air embolism in the pulmonary arteries [7,8,10]. Since ultrasound imaging was performed 8 to 10 days after the acute event, it is conceivable that any embolism-related increase in RAP would have by then subsided to the pre-embolism level. Bedside transthoracic echocardiography could have enabled the direct visualization of the heart and large vessels (and of the air bubbles trapped therein), as well as an estimation of RAP and pulmonary artery pressure. This would have provided a real-time diagnosis of air embolism and an elucidation of the underlying haemodynamic mechanisms. Unfortunately, urgent echocardiography was not accessible in the non-critical-care setting of our geriatric-medicine ward.

The cortical location of the ischaemic lesion was consistent with the embolic nature of the stroke. In fact, the surface tension of air bubbles is inversely correlated with their diameter, so that the smaller bubbles are more resistant to rupture and are thus more likely to become entrapped in the small end-arteries supplying the cerebral cortex [22]. Likewise, the preferential involvement of the frontal cortex in CAE has been attributed to the fact that it lies at the border-zone between the anterior and middle cerebral arteries, where a low flow velocity impedes the clearance of emboli [22]. Also, most lesions affect the right hemisphere [10,12,13] probably because, as they follow the bloodstream, the air bubbles entering the aorta will first encounter its first major branch, the right brachiocephalic artery, and then proceed on to the right-carotid circulation [10,12].

The clinical presentation was in keeping with the sparse case reports and case series in the literature on CAE and ischaemic “hand-knob” stroke, in which the most frequently described symptoms are impaired consciousness [7,10,11,12,13] and predominantly left-sided focal motor deficits [10,12,13,15,16,17,19]. Sensory symptoms, i.e., hand hypo- or hyper-aesthesia, are also relatively common [16,18,20] and mostly occur in the absence of sensory deficits on neurological examination. They are believed to reflect subjective sensory disturbances secondary to motor dysfunction [16].

The fact that signs of CAE were not found on brain CT deserves some discussion. The radiologic appearance of air bubbles is that of hypodense areas, circular-shaped and within the brain parenchyma in the case of arterial CAE, or serpiginous and along the cortical sulci in the case of venous CAE [22]. However, it is acknowledged that air bubbles are rapidly reabsorbed, so that their absence on brain imaging does not preclude a diagnosis of CAE [7]. In our case, brain CT, which was performed almost one hour after the onset of hand paresis, may have been unable to detect cerebral air. Nonetheless, it should be emphasized that the potential of air bubbles to produce ischaemic damage is not at odds with their evanescent nature. Indeed, cerebral hypoperfusion persists even after the disappearance of the bubbles because it is caused not only by mechanical obstruction, but also by the activation of an inflammatory response [7,11,12].

The main treatment for CAE is the administration of normobaric or hyperbaric oxygen (HBO). The efficacy of oxygen therapy is two-fold: it promotes the oxygenation of ischaemic tissues and provides a diffusion gradient that favours the egress of nitrogen from the bubbles (“denitrogenation”), thereby reducing the size of the air emboli [7,11]. HBO has an additional mode of action: according to Boyle’s law, there is an inverse relationship between the pressure and volume of a gas, so that raising ambient pressure causes the gas bubbles to shrink [7,11]. However, HBO therapy is hampered by the limited availability of hyperbaric chambers, and there are still some unresolved issues concerning its utility. The latter include the lack of randomized controlled trials proving its benefit, the question of whether it is more effective for arterial or for venous air embolism, and the time-frame within which it is useful [7,8,11,12]. In our case, only normobaric oxygen was delivered, since the patient had limited neurological dysfunction and our hospital does not have a hyperbaric facility.

The optimal body positioning of a patient with gas embolism is a matter of controversy, with some advising a head-down left-lateral decubitus position (Durant’s manoeuvre), and others a flat supine position [7,13]. Durant’s manoeuvre traps the air bubbles within the apex of the right ventricle, away from the outflow tract of the right ventricle and the pulmonary arteries, and also prevents their retrograde ascension into the cerebral veins [7]. Nevertheless, it can promote the migration of the air emboli into the veins of the lower extremities, and it can increase the risk of cerebral oedema associated with CAE [7]. Moreover, in the case of arterial embolism, the head-down position is ineffective in preventing the air emboli from being propelled into the cerebral arteries because the buoyancy of the air bubbles is not sufficient to counteract the strength of the blood flow [7]. Also, there is evidence that the left-lateral decubitus position can increase the persistence of air within the right atrium [1], potentially favouring paradoxical embolism.

Lastly, transcatheter PFO closure in older adults is an issue worthy of being addressed. The current American Academy of Neurology guidelines [21] do not recommend PFO closure for patients aged ≥60 years because the relevant randomized controlled trials have almost exclusively enrolled younger subjects, with the exception of the DEFENSE-PFO (Device Closure Versus Medical Therapy for Cryptogenic Stroke Patients with High-Risk Patent Foramen Ovale) trial, in which 28% of the participants were aged ≥60 years [23]. Yet, very recent data suggest that PFO closure may offer greater advantage at older ages, and have sparked debate within the scientific community. A subgroup analysis of the DEFENSE-PFO trial has shown that transcatheter PFO closure versus medical therapy alone was significantly more beneficial in terms of ischaemic event recurrence only in subjects aged ≥60 years, and even more so in those aged ≥70 years [24]. Along the same lines are the results of an age-focused meta-analysis by Mazzucco et al. [25] on cohort studies involving patients with cryptogenic ischaemic events. Their two main findings relate to the risk of recurrent ischaemic stroke: in patients with PFO receiving medical treatment alone, the risk increases with mean age, and in patients with and without PFO stratified by age, the risk is greater only in subjects with PFO aged ≥65 years. The authors thus conclude that the increased risk of stroke recurrence with age appears to be causally connected to the presence of a PFO, possibly because the likelihood of right-to-left shunting may be enhanced by age-related factors such as increased PFO size and increased prevalence of venous thrombosis and of pulmonary hypertension [25]. Taken together, these novel lines of evidence would seem to open the way to randomized trials of PFO closure in older adults.

## 4. Conclusions

In conclusion, CAE is a rare but potentially fatal complication of many seemingly routine invasive medical procedures, including central venous catheterization. Its relevance to clinical practice lies in the fact that it is a treatable and largely preventable condition. It should always be suspected in the case of an acute neurological event occurring in close temporal relationship with the insertion, maintenance, and removal of a CVC. Increasing awareness of CAE as an uncommon cause of iatrogenic stroke is paramount to allow its early recognition and treatment, especially among non-intensivists, as well as to promote strict adherence to CVC manipulation protocols.

Although we report of a doubly rare case (“hand-knob” stroke due to CAE), we believe this does not detract from its educational value. The large annual volume of CVC insertions means that CVC-associated CAE, despite its low prevalence, is likely to affect a considerable number of individuals. Furthermore, even if ischaemic “hand-knob” stroke is mostly benign, other clinical presentations of CAE entail high morbidity and mortality.

## Figures and Tables

**Figure 1 brainsci-12-00772-f001:**
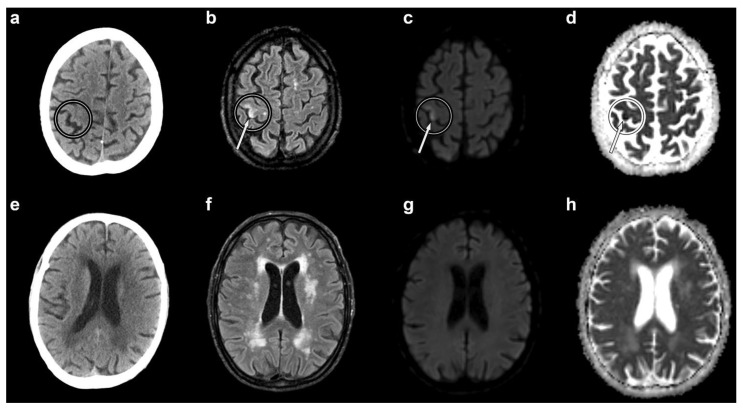
Non-contrast axial brain computed tomography (CT) and magnetic resonance imaging (MRI) scans. Circles indicate the right “hand-knob” area (horizontal epsilon-shaped). CT shows no evidence of cerebral air bubbles (**a**). The MRI fluid-attenuated recovery (**b**), diffusion-weighted (**c**), and apparent diffusion coefficient (**d**) sequences demonstrate a recent ischaemic lesion of the lateral portion of the right “hand-knob” area (arrows). Both CT and MRI show diffuse chronic cerebrovascular disease (lower panel: (**e**–**h**)).

**Figure 2 brainsci-12-00772-f002:**
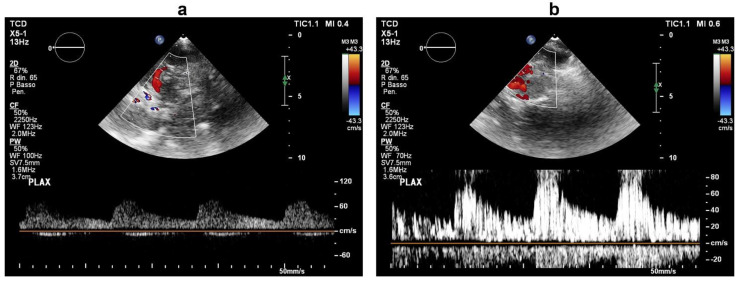
Transcranial Doppler ultrasound during intravenous infusion of agitated saline at baseline (**a**) and during a Valsalva manoeuvre (**b**). In (**b**), the middle cerebral artery Doppler signal exhibits a “shower effect” (>25 microembolic signals), which indicates a high-grade right-to-left shunt.

**Figure 3 brainsci-12-00772-f003:**
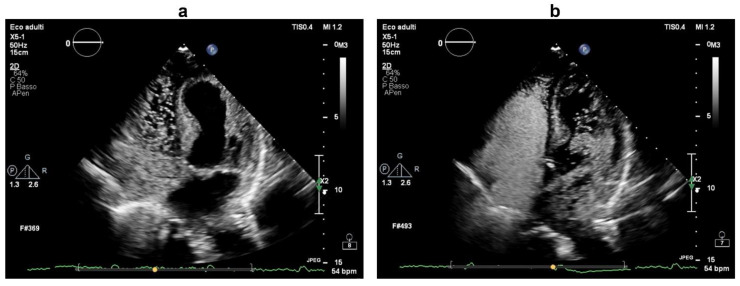
Transthoracic echocardiogram (apical four-chamber view) during intravenous infusion of agitated saline associated with a Valsalva manoeuvre. In (**a**), saline contrast can be seen filling the right heart chambers. In (**b**), numerous microbubbles (>20) can be visualized in the left heart chambers within three cardiac cycles from complete opacification of the right atrium, demonstrating a high-grade right-to-left intracardiac shunt consistent with patent foramen ovale.

## Data Availability

The data presented in this study are available in the article or Appendix A.

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
