# Peer review of "Ischaemic Stroke of the “Hand-Knob” Area Due to Paradoxical Cerebral Air Embolism after Central Venous Catheterization—A Doubly Rare Occurrence: A Case Report and an Overview of Pathophysiology, Diagnosis, and Treatment"

_brainsci, 2022, doi:10.3390/brainsci12060772_

Round 1

Reviewer 1 Report

The authors provide a great description of the case and its evaluation. But, there is an essential issue intrinsically related to the manuscript. The reviewer believes that describing such a complex occurrence (Hand-knob stroke due to CAE) does not provide something new to science.

Reviewer 2 Report

I have enjoyed reading the case report and the review. This is a well written case report and all areas have been covered. It would have been nice to have some supporting ultrasound images and loops supporting the authors speculation on the reason of air embolism in this particular patient.

Reviewer 3 Report

The manuscript entitled „Ischaemic stroke of the “hand-knob” area due to paradoxical cerebral air embolism after central venous catheterization: a doubly rare occurrence. A case report and an overview of pathophysiology, diagnosis and treatment” is a well-written article that describes an interesting case presentation of a rare case of cerebral air embolism (CAE) involving „hand-knob” area due to paradoxical mechanism in a  patient with undiagnosed patent foramen ovale (PFO).

The introduction is well-written and gives an overview of the pathophysiology of CAE mechanisms. In a case presentation section, the authors present a detailed description of the patient medical history before, during, and after the occurrence of CAE. The most exciting part of the article is Video 1, which confirms the diagnosis of a high-grade right-to-left intracardiac shunt.

The discussion section is the most abundant part of the article, yet it presents a lot of essential information that may be improved. Please shrink or move general information about PFO to the introduction section. Please remove the word „Discussion” from line 246 as well as lines 247-250.

Minor complaints:

The description below figure 1 contains many repetitions in the text. Please correct. Figure 2 description: please add a dot after „iv”

The paper would benefit from stylistic changes to the way it has been written for a more robust, clearer, and more compelling argument. 

Round 2

Reviewer 1 Report

Satisfactory.